# Preparation, Properties and Cell Biocompatibility of Room Temperature LCST-Hydrogels Based on Thermoresponsive PEO Stars

**DOI:** 10.3390/gels7030084

**Published:** 2021-07-06

**Authors:** Bagus Santoso, Paul R. Turner, Lyall R. Hanton, Stephen C. Moratti

**Affiliations:** Department of Chemistry, University of Otago, Dunedin 9016, New Zealand; y.bagus.santoso@gmail.com (B.S.); paul.turner@otago.ac.nz (P.R.T.); lhanton@chemistry.otago.ac.nz (L.R.H.)

**Keywords:** star polymer, room temperature gel, ATRP, lower critical solution temperature, cytotoxicity

## Abstract

A series of star and linear polymers based on a poly(ethylene oxide) core and poly(diethylene glycol ethyl ether acrylate) outer arms were synthesised by atom-transfer radical polymerization. The polydispersity of the polymers were low, showing good control of initiation and growth. They all showed lower critical solution (LCST) behaviour, and at 30% concentration most gelled at or below room temperature. The behaviour depended on the number and length of the arms, with the polymers with longer arms gelling at a lower temperature and producing stiffer gels. The shear modulus of the gels varied between 1 and 48 kPa, with the gelling temperature varying between 16 and 23 °C. Attempted cell cultures with the polymers proved unsuccessful, which was determined to be due to the high concentration of polymers needed for gelling.

## 1. Introduction

The word “hydrogel” is used to describe a three-dimensional network of cross-linked polymers. The spaces within these polymer matrices can be filled with water molecules and thus allow the dry material to swell many folds. The development of hydrogels as a commercial product was pioneered by Wichterle and Lim in 1960 when they described the formation of rubber-like soft gels following the free radical copolymerisation of ethylene glycol methacrylate (HEMA) and ethylene glycol dimethacrylate [1]. The material showed no irritating reaction to living organisms and was later developed into contact lenses, one of the most widely used applications of hydrogels today. Hydrogels are a highly versatile class of material and have found applications in a variety of fields including agriculture [2], textiles [3], food [4], electrochemistry [5] and engineering [6]. The most extensive use of hydrogels is in the biomedical field [7]. One reason is that they can possess physical properties similar to tissues, due to their high water content and soft consistency [8]. In the biomedical field, they have been used commercially in wound dressings [9], adhesives [6], breast implants [10], drug delivery [11,12], biosensors [13], tissue engineering [14] and hygiene products [15]. The versatility of the material class comes from the ability to tune the physical and chemical properties of the resulting material depending on the type of monomers used and the amount of cross-linking density. Certain types of hydrogels can undergo physical changes in response to an external stimulus; thus, they have been called “soft machines” [16]. These stimuli can be pH, temperature, chemicals, light, electric field or shear stress [17]. In particular, temperature-responsive gels have been very popular and have been used in biomedical applications such as drug delivery [18,19,20], tissue engineering [21,22,23], and intelligent surfaces [24,25,26,27].

In the field of thermoresponsive hydrogels, there is an emphasis on a response near the physiological temperature of 37.5 °C [7,28,29,30]. This is usually done by exploiting the lower critical solution transition (LCST) properties of certain polymers [7,31]. This allows for the hydrogel to be injected as a liquid at room temperature and then solidify to form a gel in situ. This was demonstrated by Bae et al., who reported a polyethylene oxide and poly(L-lactic acid) triblock copolymer (PEO-PLLA-PEO) which was able to be injected subcutaneously into mice [19]. Once injected, the polymer formed a gel which was able to release its contents slowly over 12 days. More recently, Li et al., reported the potential use of poly(N-isopropylacrylamide) (PNIMPAM) grafted onto alginate (alginate-*g*-PNIPAM) in cancer treatment [32]. An in vitro study demonstrated that a slow sustained release of the chemotherapy drug doxorubicin (DOX) was achieved after it was encapsulated in the hydrogels. Cellular uptake of the DOX released from the hydrogel was found to be increased. In addition, the pure alginate-*g*-PNIPAAm copolymers were noncytotoxic to the cells.

This type of material has also shown potential use in tissue engineering. Marra et al., grafted PNIPAM to a hyaluronic acid backbone [22]. The resulting gel was able to encapsulate human adipose stem cell (ASC) which survive up to 28 days of culture. Preliminary in vivo study in mice shows the maintenance of the original shape 5 days after implantation. Jung et al., designed composite hydrogels that consisted of a Pluronic polymer and cross-linked hyaluronic acids which showed gelation at body temperatures [23]. The cell viability of human ASCs in the hydrogels was about 50% after in vitro culture for 3 days. The ASCs/hydrogel mixtures were injected into mice subcutaneously, and the retrieved constructs showed that ASCs were dispersed through the hydrogel matrix.

However, for certain external applications (e.g., wound dressings or 3D printing), it might be desirable to have a biocompatible material that sets at a lower temperature than body temperature. In 3-D printing, it would mean that a heated stage was not necessary and that the constructs were stable for transport, storage, etc., at room temperature. Having gels with a range of LCSTs would mean that complicated constructs could be fabricated and one part easily removed by changing the temperature. For instance, this might be advantageous in constructing vasculature. There seems to be only one other LCST system reported in the literature that sets below 25 °C [33]. In this work, we present a series of amphiphilic star-shaped polymers that consists of a hydrophilic poly(ethylene oxide) (PEO) core and a thermoresponsive block made of poly(diethylene glycol ethyl ether acrylate) (PDEGA). When heated above the LCST of PDEGA (9–17 °C depending on molecular weight [34,35,36,37,38]), the thermoresponsive block becomes hydrophobic. At a high enough concentration, the polymer will form hydrophobic domains aggregating with neighbouring chains via physical crosslinking to form a gel [39].

## 2. Results and Discussion

### 2.1. Synthesis

There should be at least two thermoresponsive blocks on each polymer chain for a connected network to form at reasonable concentrations. To achieve this, we used poly(ethylene oxide) (PEG) star polymers as the water-soluble core, as well as simple PEG linear chain. The block-stars might have an advantage over analogous block-linear polymers. The higher concentration of end chains and the pre-existing “crosslink” in the centre of a star vs the linear polymer might allow for a lower gelation concentration, although this will be offset by the smaller length of each arm for any given molecular weight. To this end, four different molecular weight PEO-stars were used, as well as a linear PEO chain, to look at the effect of arm length and number.

There are several possible thermoresponsive blocks that could be used that are known to set at room temperature or below [31]. We settled on poly(diethylene glycol diethyl ether acrylate) as it seemed it could be polymerised by atom-transfer radical polymerization (ATRP) [40,41]. This technique allows control of molecular weights and polydispersity, as well as allowing for efficient initiation and chain growth from the end of each polymer arm. The overall synthesis is seen in Figure 1.

The poly(ethylene oxide) (PEO) starting materials were purchased from JenKem Technology and were used without further purification. These were 4-arm 10k PEO star (PEO_4_-10k) and 8-arm 10k, 20k and 40k stars (PEO_8_-10k, PEO_8_-20k, PEO_8_-40k). As a comparison, an 8k molecular weight linear PEO (PEO_2_-8k) was also used. The end groups of these star polymers were modified into ATRP initiators using a similar method as used in a previously reported synthesis of 8 arm PEO macro cross-linker [42]. By using these relatively harsh conditions, we ensured nearly complete conversion of the –OH groups as measured by NMR analysis. This was important as it is impossible to separate any partially reacted starting material. The degree of conversion was estimated from the ^1^H NMR spectrum. Under these conditions, the polymers typically reached > 95% conversion.

The ATRP of di(ethylene glycol) ethyl ether acrylate (DEGA) using PEO star initiators was carried out in anisole, using similar conditions previously reported for oligo ethylene glycol methacrylates [40]. Using this method, polymers with the desired degree of polymerisation (DP) were obtained. However, this method was not able to produce the required DP using the PEO_2_-8k initiator as the polymerisation did not proceed to complete conversion. This polymer was instead synthesised using a supplemental activation reducing agent (SARA) ATRP in DMSO [41]. Interestingly, SARA-ATRP was found to be unsuitable for polymerisation with the star initiators as it led to a cross-linked product. Table 1 summarises the molecular weights and rheometry data of the synthesised polymers. The molecular weight of the PDEGA component was estimated from the ^1^H NMR (Figure 2) of the purified polymer using the formula:(1)$$M_{n_{\mathrm{PDEGA}}}=\frac{1}{\left(I_{3.51}-8\right)/4}\times \frac{M_{w_{\mathrm{Initiator}}}}{44.05}\times 188.2$$
where $M_{n_{\mathrm{PDEGA}}}$ is the molecular weight of the PDEGA chain $M_{w_{\mathrm{Initiator}}}$ is the molecular weight of the PEO initiator and $I_{3.51}$ is the integration of the peaks between 3.38–3.62 ppm, assuming the integration of the broad peak at 4.11 ppm is 2.

For polymers PEO_4_-10k (P2) PEO_8_-10k (P3) and PEO_8_-20k (P4), the degree of polymerization (DP) of the PDEGA arm was aimed to be the same as the PEO arm. The actual results as measured by NMR are (DP PEO, DP PDEGA) P2 (57, 55), P3 (28, 40), P4 (57, 61). For the longer PEO_8_-40k (P5), the target DP of the PDEGA was aimed to be the same as done for P2 and P4. It actually ended up a little higher with a DP of 82 for the PDEGA versus 114 for the PEO. The linear PEO_2_-8k initiator (P1) was added to complete the series, and since the length of PEO per arm is slightly lower than P2, the target molecular weight of PDEGA was adjusted to compensate. The desired molecular weight of PDEGA was able to be synthesised in all the polymers. GPC measurements of the polymers showed low PDI for all polymers, suggesting good initiation and control of the polymerization of the second block. The IR spectra of the polymers showed the expected carbonyl stretch of the acrylate at 1733 cm^−1^ and a strong C-O stretch at 1097 cm^−1^ (see Appendix A for NMR and IR files).

### 2.2. Gelation Studies

The main mechanism for gelation of these type of polymers is the association of hydrophobic chain ends to form a connected network [39]. For linear triblock ABA polymers (A = water-soluble block and B = thermo-responsive block), the gelation concentration and temperature depend on the molecular weight [43]. Higher molecular weight polymers gel at a lower concentration and lower temperature. However, very few studies have been done on star polymers. The advantage of star polymers is the lower critical micelle concentration which should translate to lower critical gelation concentration [44]. There may also be a relationship between modulus and molecular weight or number of arms.

The resulting polymers had a slight green colouration, suggesting a small amount of residual copper remained even after dialysis. Each polymer was made into a 30 wt% solution in DI water and was stirred in an ice bath for 4 h before equilibrating in a refrigerator overnight. Once the gels were taken out of the refrigerator it typically took 5 min before they solidified at room temperature as determined by a vial inversion test (Figure 3). Polymers P2 (PEO_4_-10k-PDEGA) and P3 (PEO_8_-10k-PDEGA) did not pass this test and formed a very viscous fluid instead. Polymers P1 (PEO_2_-8k-PDEGA), P4 (PEO_8_-20k-PDEGA) and P5 (PEO_8_-40k-PDEGA) all formed a soft gel with a consistency similar to that of petroleum jelly. Upon deformation, P4 flowed back into its original shape overnight at 30 wt%, while P1 and P5 did not flow back. The gels were then taken out of the vial and placed in excess DI water at room temperature to observe their swelling and degradation behaviour. Gels made from P4 completely dissolved within 3 h, while P1 and P5 gels only dissolved after 12 h with minimal swelling observed. At 20 wt% concentration only P5 solidified to form a hydrogel at room temperature, the other polymers stayed as a viscous liquid.

To measure the behaviour of the gels with increasing temperatures, rheological measurements were carried at 30% *w*/*v* out using an oscillatory temperature step program. The gels were subjected to a temperature increase of 0.5 °C, allowed to equilibrate for 180 s and then measurements were taken at an oscillatory stress of 10 Pa and frequency of 1 Hz. As opposed to a simple temperature ramp, this allows the gel some time to equilibrate at the temperature measured and should give a more accurate sol–gel transition temperature. Storage (G′, red) and Loss (G″, blue) moduli of the gels between 5–30 °C are shown in Figure 4.

In general, all of the rheology results showed similar trends: both the G′ and G″ values were initially very low in value, with the G″ value higher than the G′, as expected for a liquid. As the gel was heated these values increased steadily, eventually G′ becomes higher than the G″, finally reaching a plateau. Polymer P3 (PEO_8_-10k-PDEGA) at 30% concentration did not appear to have gelled according to the rheological data. This matched our visual observation, as that gel appeared to be a viscous liquid during the vial inversion test. We suspect that for this molecular weight the polymer is not of sufficient concentration to gel. Similar behaviour was reported in a linear BAB triblock polymer (B = thermoresponsive block, A = water-soluble block) where the molecular weight of the A block was not sufficient for gelation [45].

In general, the polymers with the higher molecular weight per arm gave higher plateau moduli. Despite P2 (PEO_4_-10k-PDEGA), P3 (PEO_8_-10k-PDEGA) and P4 (PEO_8_-20k-PDEGA) being overall higher in molecular weight than P1 (PEO_2_-8k-PDEGA), the molecular weight per arm or end-to-end distance of these star polymers are significantly lower than P1. Additionally, P5 (PEO_8_-40k-PDEGA), which has a larger molecular weight per arm, showed slightly lower modulus than P1 which suggests that stars produce less stiff gels than linear polymers of the same size. The maximum shear modulus (both G′ and G″) of P4 is roughly 20 times the modulus of P2. This suggests that for star polymers with the same molecular weight per arm, increasing the number of arms greatly increases the stiffness.

Another clear trend is that the higher the final modulus, the lower the gelation temperature. This makes intuitive sense, as both would be controlled by the same factor—the number of effective crosslinks. A more efficiently cross-linkable polymer would likely form the critical number of cross-links needed for gelation earlier in the process, as the desolvation of the PDEGA block is going to be spread over a temperature range, rather than at just a very sharp value.

One advantage of star polymers over linear polymers may be the lower critical gelation concentration [44]. This was also observed here as only P5 formed a gel at 20% concentration whereas the others needed a 30% concentration. Apart from this, there seems little other reason for using a star over a linear architecture from these results, especially given the extra cost and synthetic difficulty in making the stars.

### 2.3. Cell Studies

An attempt was made to grow T0523 cells, an immortalized mesenchymal stromal cell line in the presence of a 20% *w*/*v* concentration of PEO8-40k-PDEGA as a precursor to printing gel-cell structures. However, this was unsuccessful, even though the polymer components (mainly PEO) were presumed to be non-cytotoxic. It was thought that perhaps the problem was the high concentration used, so the precursor PEO stars were investigated with the same cell line. PEO is known to be non-cytotoxic at low concentrations [46,47], but the effect of higher concentrations has not been reported. The 5 PEO cores used in the present study were incubated at 7.5%, 15%, and 30% *w*/*v* concentrations, with the resulting cell counts shown in Figure 5.

At all concentrations above 7.5% *w*/*v*, there was significant inhibition of cell survival. There seemed little correlation with the molecular weight of the PEO, though the eight-armed stars may be less cytotoxic at lower concentrations. This effect is presumably due to the osmotic pressure of the polymers causing stress on the cells. It is clear that at the concentrations needed for gel formation, the PEO stars used here will cause problems for cell culture. The only solution would be to use much higher molecular weight stars or polymers so that gel formation is possible at lower concentrations.

## 3. Conclusions

The use of di(ethylene glycol)-ethyl ether acrylate allows the synthesis of star and linear polymers, producing thermoresponsive hydrogels that set on warming to room temperature and below. The stiffness and the setting temperature of the gel can be controlled by the number and length of arms. The best gels appear able to flow over an extended period but slowly dissolve in excess water. The gels were cytotoxic at the concentrations needed for gelling. Much higher molecular weights will be needed for future cell culture experiments in order to lower the gelling concentration.

## 4. Materials and Methods

Poly(ethylene oxide) star polymers were purchased from JenKem Technology and used as received. All other chemicals were purchased from Sigma Aldrich and used without further purification. All solvents were HPLC grade with purity of 99.9%. Sodium hydride was 50% in oil. 2-Bromoisobutyryl bromide was 99% pure. Me_6_TREN was given as 97% pure. CuBr, CuBr_2_ were 99.9% pure. Di(ethylene glycol) ethyl ether acrylate was stated to be 90% pure; however, no obvious impurities could be seen by NMR. Cu wire was activated by soaking in conc. HCl for 10 min, then rinsing with water and acetone, dabbing dry and placing under argon in the polymerisation flask.

### 4.1. Characterisation

^1^H NMR and ^13^C NMR spectra were recorded on a 400 MHz Varian spectrometer in DMSO-d_6_. Chemical shifts were reported relative to the residual CDCl_3_ (^1^H, 7.26 ppm and ^13^C, 77.16 ppm) (DMSO-d_6_ (^1^H, 2.50 ppm and ^13^C, 39.52 ppm) solvent peaks according to the δ scale. Chemical shifts were rounded to the nearest 0.01 ppm and coupling constants rounded to the nearest 0.1 Hz. Infrared (IR) spectra were recorded on a Brucker Alpha-P ATR-IR spectrometer.

GPC analysis was carried out using a PL-GPC 50 (A Varian, Inc. Company) integrated GPC system equipped with a refractive index (RI) detector, two PLgel 5 µm MIXED-C (300 × 7.5 mm) columns and PLgel 5 µm Guard (50 × 7.5 mm) column. DMF was used as eluent with a flow of 1.0 mL min^−1^ at a constant temperature of 35 °C. The samples were dissolved at a concentration of 10 mg/mL in DMF and were filtered through a 0.25 µm PTFE syringe filter. A volume of 100 μL of the sample was injected for each run. The data were analysed with Cirrus GPC software version 3.2 using PEO-PEG calibration standard in the range of 615 to 1,378,000 g/mol.

Rheological measurements were performed using a Haake RS1 rheometer (Thermo Electron) with titanium cone-plate geometry (20 mm/1° cone). The samples (30% *w*/*v*) were loaded onto the lower stationary thermostated rheometer plate at 5 °C and the upper plate was adjusted to a predefined gap size (0.052 mm gap). Silicon oil was applied around the detector to prevent water evaporation during the experiment. Samples were subjected to an oscillatory stress force of 10 Pa at a frequency of 1 Hz, which was previously determined to be within the linear viscoelastic region (LVR) of the material across the temperature range. The temperature sweeps were measured at an increment of 0.5 °C with 180 s delay between measurements.

The DSC analyses were performed on a Thermal Advantage Q2000-1687 apparatus between 0–60 °C at a heating rate of 2 °C/min.

### 4.2. Synthetic Procedures

#### 4.2.1. General Synthesis of PEO-Star ATRP Initiators

To a solution of PEO star polymer (5.0 g) in dry toluene (50 mL) was added NaH (5 equivalence per OH group) at 0 °C. The mixture was allowed to stir for 1 hr to generate the anion. 2-Bromoisobutyryl bromide (2 equivalents per OH group) was added dropwise, and the reaction was stirred at 50 °C for 72 h. Excess NaH was quenched by the addition of MeOH until no further effervescence was observed. The mixture was then filtered through celite and the solvent evaporated. The polymer was precipitated twice into cold diethyl ether (150 mL). The product was then collected via vacuum filtration, 4.8 g of product was collected as a white powder (96 % yield).

^1^H NMR (400 MHz, CDCl_3_) δ 4.37–4.24 (-CH_2_-CH_2_-OCO-), 3.86–3.40 (br, PEO backbone), 1.93 (s, (CH_3_)_2_-CBr-COO-R).

#### 4.2.2. General ATRP Procedure

In a sealed flask, CuBr (17 mg, 0.12 mmol), CuBr_2_ (8.9 mg, 0.04 mmol) and Me_6_TREN (0.4 mL, 0.16 mmol) were dissolved in anisole (1 mL) and was degassed with argon for 15 min. In a separate flask, a solution of PEO star initiator (0.4 g, 0.02 mmol) and di(ethylene glycol) ethyl ether acrylate (DEGA) (2.25 g, 12 mmol) in anisole (1 mL) was degassed with argon for 15 min. This solution was then injected into the Cu complex solution using an argon-filled syringe. The reaction was then heated to 90 °C and stirred for 4 h. The solution was then diluted with ethanol and dialysed against DMF for 1 day then against DI water in the refrigerator for 2 days. Thus, 1.1g of polymer was obtained as a solid.

^1^H NMR (500 MHz, DMSO-*d*_6_) δ 4.17–4.01 (br, OCO-CH_2_-CH_2_-O-), 3.62–3.48 (br, PEO backbone, PDEGA sidechain), 3.43–3.38 (br, CH_3_-CH_2_-O-), 2.36–2.17 (br, PDEGA backbone), 1.84–1.37 (br, PDEGA backbone), 1.13–1.00 (br, CH_3_-CH_2_-O-).

#### 4.2.3. SARA ATRP Procedure

In a sealed flask, CuBr_2_ (11 mg, 0.05 mmol), ME_6_TREN (30 µL, 0.1 mmol), PEO_2_-8k-Br initiator (0.20 g, 0.1 mmol) and di(ethylene glycol) ethyl ether acrylate (DEGA) (1.88 g, 10 mmol) was dissolved in DMSO (2 mL) and degassed with argon for 15 min. The solution was injected into a sealed flask containing activated Cu wire under an argon atmosphere using an argon-filled syringe. The reaction was heated to 60 °C and stirred for 2 h. The polymer was then diluted with THF (10 mL) and dialysed against THF for 1 day followed by DI water for 2 days in a 12k MWCO dialysis bag. After lyophilisation, 0.2 g of product was obtained as a solid

^1^H NMR (500 MHz, DMSO-*d*_6_) δ 4.17–4.01 (br, OCO-CH_2_-CH_2_-O-), 3.62–3.48 (br, PEO backbone, PDEGA sidechain), 3.43–3.38 (br, CH_3_-CH_2_-O-), 2.36–2.17 (br, PDEGA backbone), 1.84–1.37 (br, PDEGA backbone), 1.13–1.00 (br, CH_3_-CH_2_-O-).

#### 4.2.4. Cell Studies

T0523 cells (1600) were placed into each well of a 96 well plate along with 30, 15 or 7.5% (*w*/*v*) PEO stars Dulbecco’s Modified Eagle Medium (DMEM) with 5 % human platelet lysate. However, at this stage, it was noticed that PEO was not soluble in the media, despite being soluble in up to 50 % (*w*/*v*) in water. To ensure solubility, 5% *w*/*v* ethanol was added. Plates were incubated at 37 °C in a tissue culture incubator with humidified 5% CO_2_ atmosphere for 7 days. Medium and compounds were then removed and the cells fixed using 4% paraformaldehyde at room temperature for 30 min. After a wash with PBS, cell nuclei were stained with 1 µg/mL Hoechst 33,342 for 1 h before visualization and photography on an Olympus IX71 inverted fluorescence microscope. Nuclei were counted from images and averages calculated from triplicate wells.

## Figures and Tables

**Figure 1 gels-07-00084-f001:**
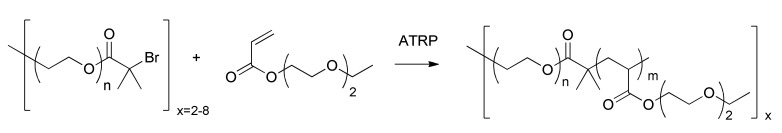
Synthesis of the thermoresponsive block polymers.

**Figure 2 gels-07-00084-f002:**
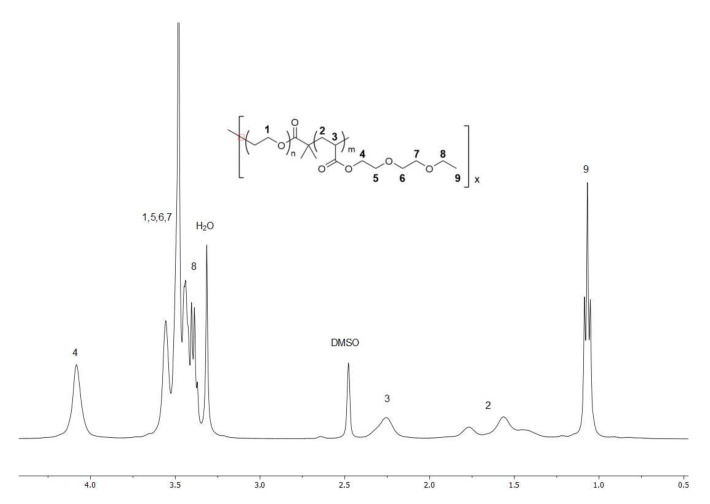
Typical ^1^H NMR of a star polymer in deuterated-DMSO.

**Figure 3 gels-07-00084-f003:**
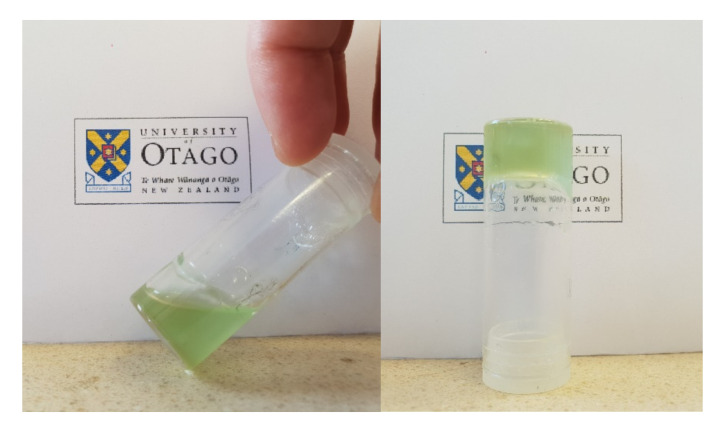
Gelation of a 30% solution of PEO_8_-40k-PDEGA (P5) on warming to RT from 0 °C.

**Figure 4 gels-07-00084-f004:**
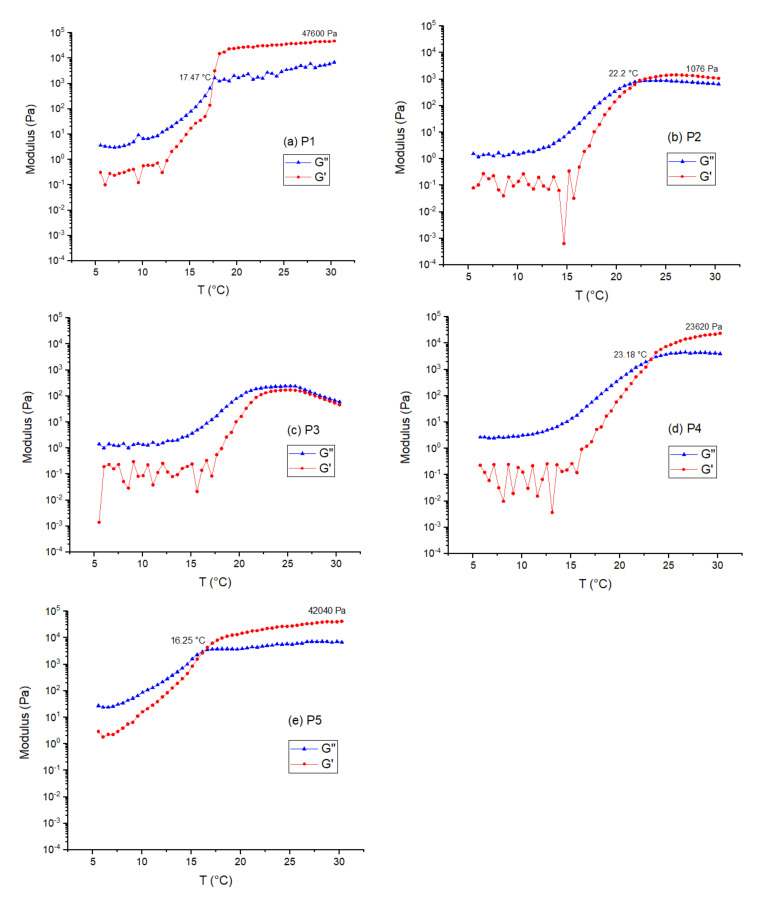
Rheometry graphs of (**a**) PEO_2_-8k-PDEGA (P1); (**b**) PEO_4_-10k-PDEGA (P2); (**c**) PEO_8_-10k-PDEGA (P3); (**d**) PEO_8_-20k-PDEGA (P4); (**e**) PEO_8_-40k-PDEGA (P5). The red line represents the storage modulus (G′); the blue line represents the loss modulus (G″).

**Figure 5 gels-07-00084-f005:**
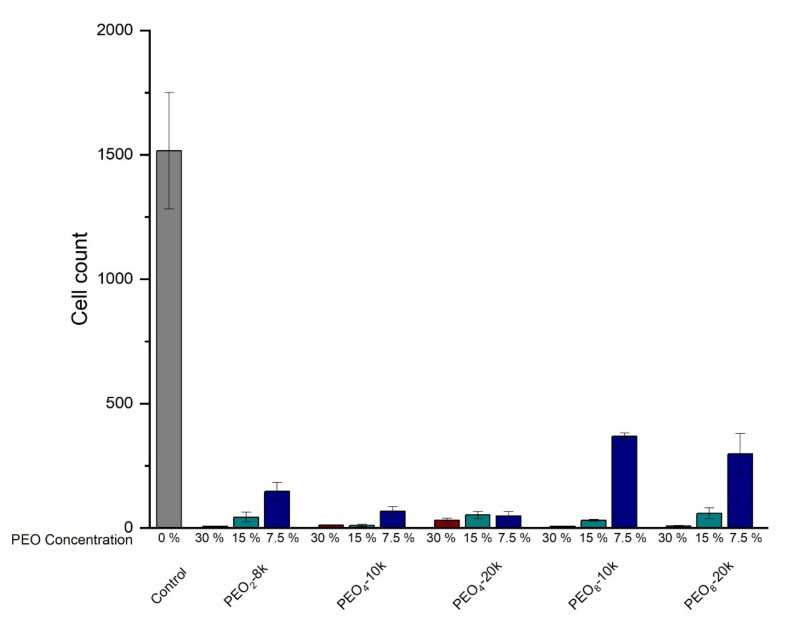
T0523 Cell survival in the presence of various concentrations of PEO stars (3 replicants).

**Table 1 gels-07-00084-t001:** Summary of the properties of the PEO-*b*-PDEGA star polymers.

			NMR ^a^		GPC ^b^		Rheometry ^c^		
Polymer	Initiator	No. of Arms	M_w_ of PEGDA	Total M_w_	M_n_ (Da)	M_w_ (Da)	G′_max_ (kPa)	G″_max_ (kPa)	T_gel_ (°C)
P1	PEO_2_-8k	2	15,700	23,600	14,500	15,300	47.5	6.0	12.2
P2	PEO_4_-10k	4	41,700	51,700	20,600	23,900	1.1	0.68	13.6
P3	PEO_8_-10k	8	50,700	60,700	23,500	26,900	0.17	0.24	-
P4	PEO_8_-20k	8	72,700	92,700	27,000	29,600	23.6	3.9	12.6
P5	PEO_8_-40k	8	84,500	124,500	40,400	41,500	42.0	7.2	11.8

^a^ In DMSO-d6. ^b^ In DMF relative to PEO/PEG standards. ^c^ Using a temperature step method, τ = 10 Pa, f = 1 Hz, concentration 30% *w*/*v*.

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
