# Peer review of "Preparation, Properties and Cell Biocompatibility of Room Temperature LCST-Hydrogels Based on Thermoresponsive PEO Stars"

_gels, 2021, doi:10.3390/gels7030084_

Round 1
Reviewer 1 Report
I read the manuscript entitled: “Preparation, Properties and Cell Biocompatibility of Room 2 Temperature LCST-hydrogels based on Thermoresponsive PEO 3 Stars”, which I think can be published in this journal. I think the discussion is good enough, but the quality of the figures must be improved. On the other hand, please add a materials part describing the purity and the companies for all the reagents. Please, I think you should add the NMR spectra associated with all the samples, and also FTIR. Those results could show differences between structures, specifically for improving the discussion of the rheological results.
Author Response
The figures were a bit low resolution and so have been increased (I presume that was what was intended). All the IR and NMR of the polymers have been included in a supplementary. We have now added a line mentioning the IR results. The NMR results are used extensively in determining the size and composition of each arm, which is mentioned many times in the discussion. The purity of all the reagents has been added.
Reviewer 2 Report
Bagus Santoso et al. have synthesized a series of star and linear polymers based on poly(ethylene oxide) core and poly(diethylene glycol ethyl ether acrylate) outer arms by atom transfer radical polymerization. These polymers are biocompatible and can have an LCST below 25 °C so that they can generate thermally responsive and biocompatible hydrogels at low temperature. The polymers all showed LCST behavior, and at 30% concentration, most gelled below room temperature. The gelation temperature and shear modulus of a gel depended on the number and length of arms. Attempts at cell culture using the polymer proved unsuccessful, attributed to the high concentration of polymer required for gelation. Much higher molecular weights of polymers will be required for future cell culture experiments to lower the gelation concentration.
The authors controlled the molecular weight of hydrophilic and thermally reactive blocks and discussed the gel formation temperature and shear modulus depending on the structure of the polymer. However, there is not enough explanation for developing polymers with LCST lower than room temperature. I don't know what advantages there are to star polymers because the physical properties of star polymers are generally lower than those of linear polymers. In addition, the gels were cytotoxic at the concentrations needed for gelling. Overall, the explanation for the comparisons and analyzes were poorly readable. Therefore, it is recommended that this paper be resubmitted after undergoing additional revision procedures.
- It seems that the English sentence was written immaturely. Please refine your English expression a little more. There are a lot of unnecessary commas and repeated expressions. Therefore, we recommend using English editing services.
- The authors note that certain external applications (such as wound dressing or 3D printing) may be desirable to use biocompatible materials to set the LCST at a lower temperature. The author should more specifically explain the need for biocompatible polymers with low LCST. For example, LCST can be easily lowered by copolymerization of hydrophobic polymers and lowered by using a salt solution such as NaCl. Please explain in detail why we need to develop low LCST biocompatible polymers.
- Please indicate the concentration of polymer solution used for rheological measurement.
- For the polymers PEO4-10k (P2) PEO8-10k (P3) and PEO8-20k (P4), it is said that the degree of polymerization of PDEGA was aimed to be the same as the PEO arms. Did you try to match the DP of the PEO arm with the DP of the PDEGA? If so, give the DP of the initiator PEO arm and the DP of PEGDA. For the longer 123 PEO8-40k (P5), it is difficult to understand that the target DP should be equal to P2 and P4. The composition of the polymer series needs further explanation.
- In page 5, “The maximum shear modulus (both G’ and G’’) of P4 is roughly 20 times the modulus of P2. This suggests that for star polymers with the same molecular weight per arm, increasing the number of arms greatly increases the stiffness. However, the advantage of star polymers over linear polymers may be the lower critical gelation concentration. This was also observed here as only P5 formed a gel at 20 % concentration. However, there seems little other advantage in using a star over a linear architecture from these results.” This paragraph is very poorly readable. I don't know what the authors want to emphasize.
Author Response
We thank the referee for his comments and their help in improving the document. The manuscript was rechecked for grammar, and we have removed many commas.
We have added a couple of sentences to exemplify the possible use of these stars as asked by the referee. “However, for certain external applications (e.g. wound dressings or 3D printing) it might be desirable to have a biocompatible material that sets at a lower temperature than body temperature. In 3-D printing it would mean that a heated stage was not necessary and that the constructs were stable for transport, storage etc at room temperature. Having gels with a range of LCSTs would mean that complicated constructs could be fabricated, and one part easily removed by changing the temperature. For instance this might be advantageous in constructing vasculature.”
The concentration has been added for the rheometry (30% w/v)
The discussion on molecular weights was a little confusing so has been clarified “For polymers PEO4-10k (P2) PEO8-10k (P3) and PEO8-20k (P4) the degree of polymerization (DP) of the PDEGA arm was aimed to be the same as the PEO arm. The actual results as measured by NMR are (DP PEO, DP PDEGA) were P2 (57, 55), P3 (28, 40), P4 (57, 61). For the longer PEO8-40k (P5) the target DP of the PDEGA was aimed to be the same as done for P2 and P4. It actually ended up a little higher with a DP of 82 for the PDEGA versus 114 for the PEO.”
P5. The paragraph included two separate observations which did not really go together. They were split, and the paragraph changed to “One advantage of star polymers over linear polymers may be the lower critical gelation concentration.[44] This was also observed here as only P5 formed a gel at 20 % concentration whereas the others needed a 30% concentration. Apart from this, there seems little other reason for using a star over a linear architecture from these results, especially given the extra cost and synthetic difficulty in making the stars.”
Round 2
Reviewer 2 Report
The authors have done a very nice job at addressing previous reviewer concerns on experimental issues and clarifications.